# The Impact of the Synergistic Effect of Temperature and Air Pollutants on Chronic Lung Diseases in Subtropical Taiwan

**DOI:** 10.3390/jpm11080819

**Published:** 2021-08-21

**Authors:** Da-Wei Wu, Szu-Chia Chen, Hung-Pin Tu, Chih-Wen Wang, Chih-Hsing Hung, Huang-Chi Chen, Tzu-Yu Kuo, Chen-Feng Wang, Bo-Cheng Lai, Pei-Shih Chen, Chao-Hung Kuo

**Affiliations:** 1Doctoral Degree Program, Department of Public Health, College of Health Sciences, Kaohsiung Medical University, Kaohsiung 807, Taiwan; u8900030@yahoo.com.tw; 2Department of Internal Medicine, Kaohsiung Municipal Siaogang Hospital, Kaohsiung Medical University, Kaohsiung 812, Taiwan; scarchenone@yahoo.com.tw (S.-C.C.); chinwin.wang@gmail.com (C.-W.W.); huangchichen@gmail.com (H.-C.C.); amorfati999@gmail.com (T.-Y.K.); kjh88kmu@gmail.com (C.-H.K.); 3Division of Pulmonary and Critical Care Medicine, Department of Internal Medicine, Kaohsiung Medical University Hospital, Kaohsiung Medical University, Kaohsiung 807, Taiwan; 4Research Center for Environmental Medicine, Kaohsiung Medical University, Kaohsiung 807, Taiwan; pedhung@gmail.com; 5Division of Nephrology, Department of Internal Medicine, Kaohsiung Medical University Hospital, Kaohsiung Medical University, Kaohsiung 807, Taiwan; 6Faculty of Medicine, College of Medicine, Kaohsiung Medical University, Kaohsiung 807, Taiwan; 7Department of Public Health and Environmental Medicine, School of Medicine, College of Medicine, Kaohsiung Medical University, Kaohsiung 807, Taiwan; p915013@kmu.edu.tw; 8Division of Hepatobiliary, Department of Internal Medicine, Kaohsiung Medical University Hospital, Kaohsiung Medical University, Kaohsiung 807, Taiwan; 9Department of Pediatrics, Kaohsiung Medical University Hospital, Kaohsiung Medical University, Kaohsiung 807, Taiwan; 10Department of Pediatrics, Kaohsiung Municipal Siaogang Hospital, Kaohsiung Medical University, Kaohsiung 807, Taiwan; 11Department of Electronics Engineering, National Yang Ming Chiao Tung University, Hsinchu 300, Taiwan; a114n.d425y@gmail.com (C.-F.W.); bclai@mail.nctu.edu.tw (B.-C.L.); 12Department of Public Health, College of Health Sciences, Kaohsiung Medical University, Kaohsiung 807, Taiwan; 13Institute of Environmental Engineering, College of Engineering, National Sun Yat-Sen University, Kaohsiung 807, Taiwan; 14Department of Medical Research, Kaohsiung Medical University Hospital, Kaohsiung 807, Taiwan; 15Division of Gastroenterology, Department of Internal Medicine, Kaohsiung Medical University Hospital, Kaohsiung Medical University, Kaohsiung 807, Taiwan

**Keywords:** temperature, air pollution, chronic lung disease

## Abstract

Previous studies have suggested an association between air pollution and lung disease. However, few studies have explored the relationship between chronic lung diseases classified by lung function and environmental parameters. This study aimed to comprehensively investigate the relationship between chronic lung diseases, air pollution, meteorological factors, and anthropometric indices. We conducted a cross-sectional study using the Taiwan Biobank and the Taiwan Air Quality Monitoring Database. A total of 2889 participants were included. We found a V/U-shaped relationship between temperature and air pollutants, with significant effects at both high and low temperatures. In addition, at lower temperatures (<24.6 °C), air pollutants including carbon monoxide (CO) (adjusted OR (aOR):1.78/Log 1 ppb, 95% CI 0.98–3.25; aOR:5.35/Log 1 ppb, 95% CI 2.88–9.94), nitrogen monoxide (NO) (aOR:1.05/ppm, 95% CI 1.01–1.09; aOR:1.11/ppm, 95% CI 1.07–1.15), nitrogen oxides (NO_x_) (aOR:1.02/ppm, 95% CI 1.00–1.05; aOR:1.06/ppm, 95% CI 1.04–1.08), and sulfur dioxide (SO_2_) (aOR:1.29/ppm, 95% CI 1.01–1.65; aOR:1.77/ppm, 95% CI 1.36–2.30) were associated with restrictive and mixed lung diseases, respectively. Exposure to CO, NO, NO_2_, NO_x_ and SO_2_ significantly affected obstructive and mixed lung disease in southern Taiwan. In conclusion, temperature and air pollution should be considered together when evaluating the impact on chronic lung diseases.

## 1. Introduction

Chronic lung diseases, including obstructive lung diseases (such as asthma, chronic obstructive pulmonary disease (COPD), and bronchiectasis), restrictive lung diseases (such as interstitial lung disease, pulmonary fibrosis and neuromuscular disease), and mixed (obstructive and restrictive) lung disease are diagnosed by spirometry according to the standardized European Respiratory Society/American Thoracic Society guidelines [1]. The United Kingdom Biobank study showed that higher exposure to various air pollutants was significantly associated with lower lung function [2]. In addition, the European Study of Cohorts for Air Pollution Effects (ESCAPE) demonstrated that, even at very low levels, air pollutants had adverse effects on lung function in adults, including both forced expiratory volume in 1 s (FEV1) and forced vital capacity (FVC) [3]. Moreover, the effect estimates were stronger for FVC than FEV1 for various pollutants, suggesting greater effects on restrictive than obstructive lung diseases [4]. However, a large population-based cohort study also confirmed the relationship between air pollution and incidence of COPD [5]. Air pollution can also induce acute exacerbations of obstructive lung diseases and increase respiratory morbidity and mortality [6]. In addition, acute and chronic air pollution exposure had also been associated with an increased risk of cardiovascular and respiratory morbidity and mortality [7,8,9]. The use of medical services in patients with chronic diseases had been reported to increase with higher levels of exposure to air pollution [10], especially in industrialized regions [11]. Air pollutants include coarse particulate matter (PM_10_), fine particulate matter (PM_2.5_), carbon monoxide (CO), nitrogen monoxide (NO), nitrogen dioxide (NO_2_), nitrogen oxides (NO_x_), sulfur dioxide (SO_2_) and ozone (O_3_) can penetrate into the lung parenchyma and alveoli, and induce the production of various inflammatory mediators, such as mitogen-activated protein kinase and nuclear factor kappa-light-chain-enhancer of activated B cells (NF-κB), which can lead to various chronic lung diseases [12]. In particular, particulate matter (PM) exposure has been associated with acute hospital admissions [13], systemic oxidative damage [14], inflammation [15], and an increased risk of exacerbations and respiratory symptoms in children and adults with existing lung diseases [14,16]. A cumulative effect of PM_2.5_ and NO_2_ has been reported prior to disease exacerbations, and high levels of both pollutants have been shown to increase the probability of COPD attacks [17]. Previous studies have confirmed that long-term exposure to fine particles had an effect on airway inflammation in older women. A study reported that exposure to PM_2.5_ was associated with a higher risk of acute exacerbations of COPD, especially in females and elderly patients (age > 75 years old), and that this effect was restricted to the cold season (November to April) [18]. In addition, it is recommended that elderly patients with chronic diseases avoid strenuous exercise outdoors when the concentration of outdoor PM_2.5_ is higher than normal [19]. Furthermore, dynamic changes in air pollutants and meteorological factors coexist simultaneously [20]. Studies from both Northeast Asia and Europe reported that a synergistic effect between high temperature and air pollution may be associated with a higher risk of mortality [21,22]. Hansel et al. reported that an increased risk of mortality in patients with lung diseases (such as COPD) was related to extremely high and low temperatures [23]. In addition, Jo et al. demonstrated that decreasing relative humidity and increasing PM level were associated with significant increases in lung disease-related admissions [24]. In this study, we investigated association among air pollutants, meteorological factors, and chronic lung diseases, and explored the interactions and synergic effects between various air pollutants.

## 2. Materials and Methods

### 2.1. Data Source

We conducted this cross-sectional study using two large databanks: Taiwan Biobank (TWB) and Taiwan Air Quality Monitoring Database (TAQMD). The TWB was the largest biobank in Taiwan and was directly supported by the government. It is comprised of volunteers with no history of cancer aged between 30 and 70 years. All participants signed informed consent before being included in the TWB, after which they underwent a face-to-face holistic interview, blood sampling, physical examination, and then completed a questionnaire which addressed personal information and lifestyle factors [25,26]. The TAQMD was established by the Executive Yuan of the Taiwan Environmental Protection Administration, and is comprised of daily air pollutant concentration data from 74 air quality monitoring stations around Taiwan. TAQMD data was used to establish a connection between the participant’s residential area and the location of the nearest air quality monitoring station, then to estimate the outdoor air pollution exposure of each participant, and finally combine the TWB data for further analysis. We also calculated the average concentrations of air pollutants in a selected year, including PM_2.5_, PM_10_, CO, NO, NO_2_, NO_x_, SO_2,_ and O_3_ using the following three-step method. (1) The corresponding longitude and latitude of the residential address were determined using Google geocoding. (2) The nearest air quality monitoring station was identified from that point as an interpolation point. (3) Data from the mapped station were filtered from the survey date to the previous year, and the average value of each air pollution indicator was calculated. 

### 2.2. Collection of Demographic, Laboratory and Meteorological Factors

The following baseline variables were recorded: demographic characteristics (age, sex, smoking, alcohol drinking history), comorbidities (hypertension, type 2 diabetes, renal failure, metabolic syndrome, and coronary artery disease), anthropometric parameters (height, weight, body mass index (BMI), body adiposity indices (BAI), body roundness index (BRI)), biochemical parameters (hematocrit (Hct), albumin, glycohemoglobin (HbA1_C_), glutamate pyruvate transaminase (GPT) and creatinine), Taiwan monitoring regions (northern, central, and southern regions), and meteorological factors (temperature (°C), relative humidity (%), and rainfall (mm)).

### 2.3. Assessment of Lung Function Status 

The lung function parameters in the Taiwan Biobank include forced expiratory volume in one second (FEV1), forced vital capacity (FVC), FEV1/FVC% ratio, FVC-predicted value, and FEV1-predicted value. FVC-predicted and FEV1-predicted values were determined by dividing the measured value by the reference value, which was calculated from a formula derived from the general population based on gender, age, height, and Asian ethnicity. The spirometry measurements were performed by well-trained technicians using the MicroLab spirometer and Spida 5 software (Micro Medical Ltd., Rochester, Kent, UK) [27]. We defined chronic lung diseases in terms of three abnormal types of lung function, including obstructive lung diseases (FEV1/FVC < 70%, such as asthma, chronic obstructive pulmonary disease, and bronchiectasis), restrictive lung diseases (FEV1/FVC > 70% and FVC < 80%, such as interstitial lung disease, neuromuscular disease, and obesity), and mixed (obstructive and restrictive) lung disease, according to the standardized European Respiratory Society/American Thoracic Society guidelines [1].

### 2.4. Ethics Statement

This study was approved by the Institutional Review Board of the Affiliated Hospital of Kaohsiung Medical University (KMUHIRB-E(I)-20180242). In accordance with institutional requirements and the principles of the Declaration of Helsinki, written informed consent was obtained from each participant. The TWB has obtained ethics certification from the Review Board/IRB-BM of the Taiwan Academy of Biomedical Sciences and the Taiwan TWB Ethics and Governance Committee.

### 2.5. Statistical Analysis

Descriptive results were analyzed between spirometry groups using one-way ANOVA and chi-squared/Fisher’s exact tests, as appropriate. Cramer’s V test was used to describe the magnitude of the association between categorical variables for a contingency table larger than 2 × 2. Spearman’s rank correlation was used to measure the strength of the relationship between two continuous variables. Crude odds ratios (ORs) and 95% confidence intervals (CIs) were estimated using multinomial logistic regression. Adjusted ORs and 95% CIs were estimated using stepwise multinomial logistic regression, and were used to identify the covariant factors associated with the development of lung diseases (spirometry groups) when the factors showed a significant association in the crude analysis. Interactions between meteorological factors and air pollution factors and between monitoring area and air pollution factors were tested according to the significance of the interaction term in the multinomial logistic regression analysis. After confirming data consistency, continuous variables, such as pollution factors that were not normally distributed, were log-transformed to achieve normality before statistical analysis. All data analyses were performed using SAS software version 9.4 (SAS Institute Inc., Cary, NC, USA).

## 3. Results 

### 3.1. Descriptive Statistics of the Demographic, Laboratory, Meteorological Factors and Air Pollutants

The mean age of the 2889 enrolled participants was 50.14 ± 10.64 years. Of these participants, 1323 (45.8%) were men and 1566 (54.2%) were women. The participants were stratified into four groups according to lung function test results: control group (normal spirometry group), and three chronic lung disease groups (obstructive group, restrictive group, and mixed group). There were no obvious differences with respect to smoking and alcohol consumption among the groups. In addition, when compared with the normal spirometry group, we found that factors associated with higher risk of chronic lung diseases include elderly age (>60 years), female gender, lower body height and weight, higher body adiposity index and body roundness index, lower haematocrit, higher glycohemoglobin, and lower albumin level (Table 1). These results were confirmed by post hoc analysis (Appendix A). In addition, exposure to CO, NO, NO_2_, NO_x_ and SO_2_ in the environment increased the systemic impact on the patients with chronic lung diseases, especially in mixed lung disease (Table 1). With regards to the baseline characteristics of air pollution factors and meteorological factors, the lowest and highest average annual temperatures were 21.46 °C and 26.36 °C (average, 24.33 °C), respectively. The lowest, highest and average levels of all air pollutants and meteorological measures were presented on Appendix A.

### 3.2. Correlation between Meteorological Factors and Outdoor Air Pollutants

We found that temperature was positively correlated with PM_2.5_, PM_10_, O_3_, and SO_2_ concentrations. Relative humidity was negatively correlated with other meteorological factors and air pollutants except for temperature and O_3_ levels. PM_2.5_ concentrations was positively correlated with all meteorological factors except for relative humidity, and all air pollutants except for O_3_ and NO. Except for PM_2.5_ and PM_10_, CO had very high correlations with all air pollutants (especially NO, NO_2_, and NO_x_). In addition, there was a moderate negative correlation between CO and O_3_. (Appendix A). 

### 3.3. Associations between Chronic Lung Disease and All Factors

Appendix A shows the crude ORs of all factors in the three chronic lung disease groups compared with the normal group. In the obstructive group, more of the participants were ≥60 years old, had lower Hct levels, higher BAI and HbA1c values, and more had type 2 diabetes as a comorbidity. In the restrictive group, more of the participants were aged ≥60 years, had lower body height, higher BAI, BRI and HbA1c values, and more had type 2 diabetes. In the mixed group, more of the participants were female and aged ≥60 years, had lower body height and weight, higher BAI, and lower albumin level.

Table 2 shows the adjusted ORs of all factors. In the obstructive group, multivariate analysis identified five independent predictive factors: HbA1c (OR 1.13; 95% CI 1.01–1.27; *p* = 0.0312), southern region (OR 0.61; 95% CI 0.39–0.97; *p* = 0.0351), temperature (OR 1.58; 95% CI 1.35–1.86; *p* < 0.0001), relative humidity (OR 1.06; 95% CI 1.01–1.11; *p* = 0.0104), and SO_2_ (OR 1.26; 95% CI 1.09–1.45; *p* = 0.0015).

In the restrictive group, multivariate analysis identified seven independent predictive factors: height (OR 0.96; 95% CI 0.94–0.98; *p* < 0.0001), HbA1c (OR 1.28; 95% CI 1.09–1.51; *p* = 0.0028), central region (OR 0.32; 95% CI 0.13–0.78; *p* = 0.0124), southern region (OR 0.31; 95% CI 0.13–0.74; *p* = 0.0083), CO (OR 511.99; 95% CI 8.08–32,425.36; *p* = 0.0032), NO_2_ (OR 1.39; 95% CI 1.14–1.70; *p* = 0.0012), and NO_x_ (OR 0.76; 95% CI 0.66–0.87; *p* = 0.0001). In the mixed group, multivariate analysis identified seven independent predictive factors: age ≥60 years (OR 2.64; 95% CI 1.35–5.13; *p* < 0.0043), height (OR 0.93; 95% CI 0.91–0.96; *p* < 0.0001), central region (OR 0.07; 95% CI 0.02–0.24; *p* < 0.0001), southern region (OR 0.14; 95% CI 0.05–0.44; *p* = 0.0007), temperature (OR 1.87; 95% CI 1.28–2.77; *p* = 0.0020), relative humidity (OR 1.15; 95% CI 1.03–1.28; *p* = 0.0108), and NO_2_ (OR 1.58; 95% CI 1.25–1.99; *p* = 0.0001). Compared with the northern region, the participants in the southern and central regions seemed less likely to suffer from chronic lung diseases. However, after adjusting for temperature and monitoring region, opposite results were obtained.

### 3.4. Interactions between Temperature and Air Pollutants 

Figure 1 shows a graph of relationship among air pollutants and temperature; in most graphs, the curve of the normal group was lower than the other three chronic lung diseases groups. In the graphs of CO, NO, NO_2_, NO_x_ and SO_2_, there was a V/U-shaped curve between temperature and air pollutants. When the temperature was at the lowest or highest, the concentrations of these air pollutants were at the highest, but at around 24.3–24.9 °C (optimum temperature), their concentrations were the lowest. However, the findings for O_3_ were the opposite, and at the highest or lowest temperature, the O_3_ concentration was the lowest, whereas at around 24.9 °C the concentration was the highest. In addition, the concentrations of PM_2.5_ and PM_10_ elevated with increasing temperature.

This shows a graph of air pollutants and temperature, with the horizontal axis showing temperature and the vertical axis showing the various air pollutants. The purple curve represents the distribution of the normal population under air pollution and temperature, the pink line represents the obstructive lung disease group, the green line represents the restrictive lung disease group, and the brown line represents the mixed-type lung disease group temperature. These findings indicated that temperature may be a key factor mediating the interactions.

### 3.5. Interactions between Temperature and Monitoring Regions

Appendix A confirms the interactions between temperature and monitoring region, temperature and air pollutants, and monitoring region and air pollutants in three chronic lung disease groups. We found a moderate relationship between monitoring region and temperature by the Cramer’s V analysis on Appendix A, and the impact of air pollutants seemed to be consistent between the northern and central regions; therefore, we combined the northern and central regions into one group and compared it with the southern region group later.

### 3.6. Correlations between Temperature and Air Pollutants

We selected different turning points to divide the temperatures into higher and lower temperature groups in the different air pollutant groups (Appendix A). Due to the CO data being skewed, log transformation was carried out before further analysis. Table 3 shows the final relationships between temperature and the effect of air pollutants. Interestingly, lower temperatures had a greater effect on air pollutant concentrations, and this finding was consistent in restrictive and mixed lung disease groups. Although we found the same phenomenon at higher temperatures, the effect was not as strong as that at lower temperatures. In comparisons of the restrictive lung disease group with the normal group: for CO exposure, OR 1.78 and 95% CI 0.98–3.25; for NO exposure, OR 1.05 and 95% CI 1.01–1.09; for NO_x_ exposure, OR 1.02 and 95% CI 1.00–1.05; and for SO_2_ exposure, OR 1.29 and 95% CI 1.01–1.65. In comparisons of the mixed-type lung disease group with the normal group: for CO exposure, OR 5.35 and 95% CI 2.88–9.94; for NO exposure, OR 1.11 and 95% CI 1.07–1.15; for NO_2_ exposure, OR 1.13 and 95% CI 1.08–1.19; for NO_x_ exposure, OR 1.06 and 95% CI 1.04–1.08; and for SO_2_ exposure, OR 1.77 and 95% CI 1.36–2.30. When comparing the obstructive lung disease group with the normal group at lower temperatures, the results showed: for NO exposure, OR 1.03 and 95% CI 1.01–1.06; and for NO_x_ exposure, OR 1.01 and 95% CI 1.00–1.03.

In Table 4, we found that the southern region of Taiwan had a significant impact on air pollution in the obstructive and mixed lung disease group. However, in the northern and central region groups, we only found this association in the mixed lung disease group. In southern Taiwan, comparisons between the obstructive lung disease group and the normal group showed: for CO exposure, OR 4.17 and 95% CI 2.08–8.37; for NO exposure, OR 1.28 and 95% CI 1.15–1.44; for NO_2_ exposure, OR 1.13 and 95% CI 1.08–1.18); for NO_x_ exposure, OR 1.09 and 95% CI 1.06–1.13; and for SO_2_ exposure, OR 1.65 and 95% CI 1.45–1.89. Comparisons of the mixed-type lung disease group with the normal group showed: for CO exposure, OR 18.85 and 95% CI 3.53–100.59; for NO exposure, OR 1.33 and 95% CI 1.02–1.73; for NO_2_ exposure OR 1.25 and 95% CI 1.13–1.37; for NO_x_ exposure, OR 1.16 and 95% CI 1.08–1.25, and for SO_2_ exposure, OR 1.83 and 95% CI 1.40–2.38. Exposures to PM_2.5_, PM_10_, and O_3_ were not associated with any demographic, clinical, or spiro-metric characteristics. Comparisons of the restrictive lung disease group with the normal group showed: for CO exposure, OR 11.58 and 95% CI 3.29–40.81; for NO_2_ exposure, OR 1.09 and 95% CI 1.01–1.18; and for SO_2_ exposure, OR 1.29 and 95% CI 1.00–1.65.

## 4. Discussion

In this analysis of 2889 participants registered in the TWB, we found that factors associated with higher risk of chronic lung diseases include elderly age (>60 years), female gender, lower body height and weight, higher body adiposity index and body roundness index, lower haematocrit, higher glycohemoglobin, and lower albumin level. We also assessed the synergistic effects of temperature, monitoring area and air pollutants on chronic lung diseases. The results showed a V/U-shaped relationship between temperature and air pollutants, and both lower and higher temperatures increased the risk of air pollution, but this effect was more obvious at lower temperatures and in southern Taiwan. Finally, we found that CO, NO, NO_2_, NO_x_, and SO_2_ levels were strongly associated with chronic lung diseases, while other air pollutants (O_3_, PM_2.5_ and PM_10_) were not.

The first important finding of this study was that elderly age (>60 years), female gender, lower body height and weight, higher body adiposity index and body roundness index, lower hematocrit, higher glycohemoglobin, and lower albumin level had the higher risk for chronic lung diseases. In addition, lower body weight and height but higher BAI means that weight is low, but body adipose tissue content is high. For example, normal weight obesity (NWO) is a term used to describe patients with higher body adiposity index but normal body weight and body mass index (BMI) [28,29]. The Towards a Revolution in COPD Health (TORCH) study showed that high-risk factors for moderate and severe COPD exacerbations were older age (patients ≥75 years of age), lower body mass index (BMI < 20 kg/m^2^), and female with poor baseline lung function (females had 1.42 times higher exacerbation rate than males) [30]. A study in C57BL/6J mice showed that estrogen deficiency and an increased estrogen receptor α expression led to the development of emphysema in aging female mouse lungs, and that these conditions improved after 17β-estradiol (E2, 0.025 mg) treatment [31,32]. In a study in aging female C57BL/6J mice, the absence of E2 resulted in a decrease in hydroxyproline content, macrophage number, and respiratory chain complex-1 protein, particularly in estrogen-deficient mice exposed to cigarette smoke [33]. In addition, Tian et al. reported that the effects of PM were stronger at high temperatures in elderly women, possibly due to menopause [34]. Moreover, a systematic review showed an increased rate of lung function decline and a significant risk of chronic lung diseases in menopausal women [35]. Thus, we speculate that women may be more vulnerable than men to develop chronic lung diseases upon exposure to air pollution, and that estrogen therapy may be a potential solution for aging-related lung diseases in women.

Furthermore, we found a V/U-shaped relationship between temperature and air pollutants (Figure 1), which means that, when the temperature was around 24.3–24.9 °C (optimum temperature) (Appendix A), the concentration of air pollutants was at the lowest. Besides, both lower and higher temperatures increased the risk of air pollution, but this was more pronounced at lower temperatures. A previous study indicated that temperature may modify the effect of pollution exposure [36]. Moreover, a high temperature has been associated with an increase in COPD-related hospital admissions [37], and a decrease in survival of older patients with chronic lung diseases [38]. Another study showed that cold temperatures increased the COPD exacerbation rate [39]. Air pollution concentrations are continuously affected by temperature changes and are often associated with exacerbations of COPD in older patients [40]. Two previous studies demonstrated that both extremely high and low temperatures were associated with an increase in the incidence of lung diseases [41,42]. In the context of climate change, as temperatures rise and extreme weather events increase, people are paying more attention to the impact of the interaction between air pollution and meteorological factors.

Another important finding of the present study was that temperature, monitoring region, and air pollutants had a synergistic effect on chronic lung diseases. Compared with northern and central Taiwan, southern Taiwan has a serious air pollution problem, mainly due to heavy industrial and commercial activity, especially in Kaohsiung City [43]. Almost all air pollutants showed higher concentrations in southern Taiwan, where the temperature is also higher, and where there is a higher density of industrial factories (there are more than 5000 factories in Kaohsiung City). We found interactions between temperature and monitoring region, temperature and air pollution, and monitoring region and air pollution (Appendix A). Thus, our results showed that people living in the southern region had a higher likelihood of being affected by serious air pollution and induced chronic lung diseases (Table 4). As a result, we assume that the synergistic reactions among temperature, monitoring region, and air pollutants are risk factors for chronic lung diseases due to the further adsorption of other pollutants.

The last important finding of this study was that CO, NO, NO_2_, NO_x_, and SO_2_ levels were positively correlated with the incidence of chronic lung diseases, especially at low temperatures. However, the other air pollutants (O_3_, PM_2.5_, and PM_10_) were not correlated with the incidence of chronic lung diseases. The toxicity of CO lies in its high binding affinity for hemoglobin to produce carboxyhemoglobin, which changes the conformation of hemoglobin and reduces its ability to carry oxygen [44]. Although previous studies have shown that low concentrations of CO can be used as a treatment option for COPD, this method is still experimental and not routinely used in the clinical setting [45], and it is still considered to be an environmental air pollutant. A case-control study in Malawian adults found that chronic lung symptoms were associated with CO exposure (OR 1.46; 95% CI 1.04–2.05) [46]. Inhaled NO therapy can be used to treat adult respiratory distress syndrome, primary pulmonary hypertension, and chronic obstructive pulmonary disease [47]. However, a review in the Cochrane database reported that inhalation of nitric oxide may temporarily increase oxygenation level, but it is not statistically significant for one-month survival [48]. In addition, NO_2_ has been demonstrated to trigger an inflammatory response, including in vitro and in vivo increases in interleukin-8 (IL-8) concentrations [49]. Furthermore, increased systemic and sputum IL-8 concentrations have been associated with COPD exacerbations [50]. Ambient NO_2_ can intensify allergic inflammatory reactions in the airways without causing symptoms or pulmonary dysfunction [51]. Jiang et al. reported that, with an increase of 10 ppb in NO_2_, the predicted values of FVC and FEV1 decreased by 0.12% and 0.37%, respectively, on the second day [52]. Besides, Duan et al. revealed that with an increase of 10 ppb in NO_2_ in the cold season, the daily mortality rate increases by 4.45% within a lag time of 0 to 2 days [53]. Goudarzi et al. reported that SO_2_ is mostly produced by exhaust gas discharged from industrial production processes, and that it eventually leads to deterioration of the surrounding air quality [54]. An animal study demonstrated activation of the TLR4/p38/NF-κB pathway accompanied by excessive secretion of pro-inflammatory cytokines in the lungs of rats exposed to PM_2.5_ and SO_2_, indicating inflammatory, pathological, and ultrastructural damage [55]. Taken together, we suggest that CO, NO, NO_2_, NO_x_, and SO_2_ cause chronic lung diseases through inflammation, and that high ambient concentrations of CO, NO, NO_2_, NO_x_, and SO_2_ are possible risk factors for chronic lung diseases. The average level of O_3_ in this study was lower than the world average [56], and therefore its impact on lung diseases might not be obvious. In addition, O_3_ levels had the opposite effect to other air pollutants (such as NO_x_) in this study, which is consistent with a previous study [57]. A possible explanation is that NO_x_ and volatile organic compounds react and combine under ultraviolet light to form ground-level O_3_ [58]. Therefore, when the NO_x_ concentration drops, the O_3_ concentration will gradually increase. Although, there was no obvious relationship between PM_2.5_, PM_10_ and chronic lung disease in this study, a previous study has pointed out that the concentration of PM increases with temperature, and thus the impact of PM_2.5_ on lung diseases may only occur at higher temperatures [59]. In addition, a study showed that hospital admission rates for respiratory diseases increased with increasing PM and temperature [24]. Our data was consistent with the above study result. Furthermore, it is possible that the lack of association between PM_2.5_ and PM_10_ and the incidence of chronic lung diseases in this study may be due to the level of exposure not being high enough or the duration of exposure not being long enough.

To the best of our knowledge, this is the first study to comprehensively investigate the association between chronic lung diseases (classified by lung function) and air pollution, meteorological factors, and anthropometric indices. The novelty of this study is that we found a V/U-shaped relationship between temperature and air pollutants, which means that when the temperature is around 24.3–24.9 °C (optimum temperature), the concentration of air pollutants is at the lowest. However, there are also several limitations to this study. First, this is a cross-sectional study; therefore, we could not assess the relationship between air pollution and the progression of lung function over time or the occurrence of chronic lung diseases. A follow-up cohort study is needed to confirm the results. Second, we used lung function assessments to identify chronic lung diseases. However, we could not obtain follow-up lung function data or imaging studies to confirm the progression of disease. Third, we used the participants’ residential address as the air pollutant exposure point. However, this is a rough measurement of air pollution exposure, and it may not accurately reflect the actual personal exposure. In addition, we did not have information regarding indoor air quality. Lung function can also be influenced by exposure to indoor air pollutants (such as brominated diphenyl-ethers, polychlorinated biphenyls [60], volatile organic compounds, polycyclic aromatic hydrocarbons, PM_2.5_ (cigarette smoke, and cooking) [61]. This will be an important topic for further study. Finally, the Taiwan Biobank does not contain information regarding occupational exposure to toxic substances, and therefore this variable cannot be analyzed in this study.

In conclusion, when compared with the normal spirometry group, we found that factors associated with higher risk of chronic lung diseases include elderly age (>60 years), female gender, lower body weight and height, higher BAI and BRI, lower haematocrit, lower albumin level, and higher glycohemoglobin level (Table 1). We also observed interactions and synergistic effects among temperature, air pollutants and different monitoring regions. Furthermore, both high and low temperatures, and exposure to various air pollutants including CO, NO, NO_2_, NO_x,_ and SO_2_ have strong impacts on the development of chronic lung diseases, especially at low temperatures (<24.6 °C) and in heavy industrial regions such as southern Taiwan. This study highlights the importance of air pollution in chronic lung diseases. We suggest that temperature and air pollution should be considered together when evaluating the impact on chronic lung diseases.

## Figures and Tables

**Figure 1 jpm-11-00819-f001:**
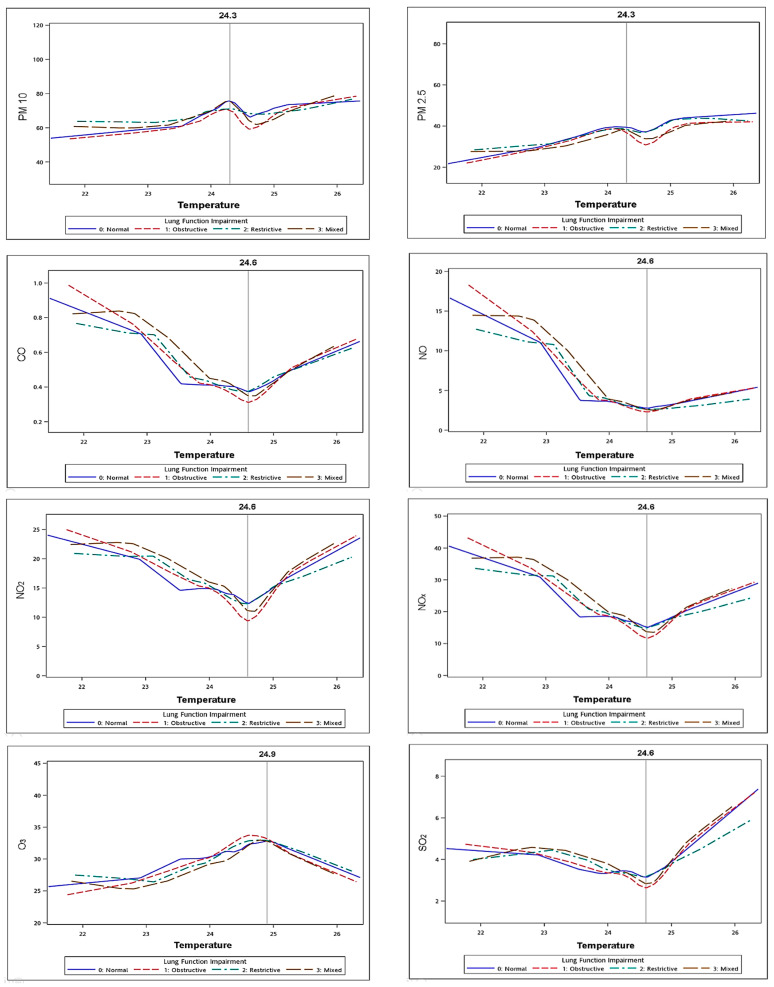
Plot of Temperature and Air pollutants in four groups.

**Table 1 jpm-11-00819-t001:** Descriptive statistics of the demographic, laboratory, meteorological factors, and air pollutants.

	Total	Normal Spirometry(1)	Obstructive Impairment(2)	Restrictive Impairment(3)	Mixed Impairment(4)	*p*-Value
*n*	2889	1902	733	154	100	
Age (years), mean (SD)	50.14 (10.64)	49.51 (10.46)	50.56(10.68)	52.61 (11.02)	55.31 (11.25)	<0.0001
30–39	628 (21.7)	438 (23.0)	149 (20.3)	26 (16.9)	15 (15.0)	
40–49	760 (26.3)	524 (27.5)	186 (25.4)	37 (24.0)	13 (13.0)	
40–59	891 (30.8)	585 (30.8)	231 (31.5)	49 (31.8)	26 (26.0)	
≥60	610 (21.1)	355 (18.7)	167 (22.8)	42 (27.3)	46 (46.0)	<0.0001
Sex, *n* (%)						

Male	1323 (45.8)	903 (47.5)	322 (43.9)	66 (42.9)	32 (32.0)	
Female	1566 (54.2)	999 (52.5)	411 (56.1)	88 (57.1)	68 (68.0)	0.0098
Smoking, *n* (%)						
None	2112 (73.1)	1382 (72.7)	535 (73)	116 (75.3)	79 (79.0)	
Current & Former	777 (26.9)	520 (27.3)	198 (27)	38 (24.7)	21 (21.0)	0.5031
Alcohol consumption, *n* (%)						
None & sometimes	2605 (90.2)	1711 (90.0)	660 (90.0)	143 (92.9)	91 (91.0)	
Quit	67 (2.3)	47 (2.5)	16 (2.2)	4 (2.6)	0 (0.0)	
Current	217 (7.5)	144 (7.6)	57 (7.8)	7 (4.5)	9 (9.0)	0.5520
Anthropometric parameter, mean (SD)						
Height (cm)	162.63 (8.26)	163.09 (8.32)	162.52 (8.05)	160.26 (7.57)	158.28 (7.80)	<0.0001
Weight (kg)	64.10 (12.10)	64.49 (12.22)	64.08 (11.6)	62.74 (12.94)	58.82 (10.67)	<0.0001
Body Mass Index mean (kg/m^2^)	24.12 (3.45)	24.13 (3.44)	24.16 (3.32)	24.34 (4.22)	23.38 (3.27)	0.1457
Body Adiposity Index	28.60 (3.92)	28.40 (3.92)	28.75 (3.79)	29.62 (4.38)	29.72 (3.79)	<0.0001
Body Roundness Index	3.72 (1.13)	3.70 (1.12)	3.74 (1.08)	3.96 (1.38)	3.70 (1.20)	0.0475
Biochemical data, mean (SD)						
Hematocrit (%)	44.15 (4.61)	44.32 (4.64)	43.86 (4.61)	44.13 (4.52)	43.10 (3.86)	0.0139
Glycohemoglobin (%)	5.78 (0.79)	5.73 (0.71)	5.84 (0.90)	6.00 (1.13)	5.82 (0.73)	<0.0001
Albumin (g/dL)	4.60 (0.24)	4.60 (0.24)	4.6 (0.24)	4.56 (0.25)	4.54 (0.24)	0.0241
Serum glutamic pyruvic Transaminase (IU/L)	24.91 (22.64)	24.84 (22.51)	24.93 (24.01)	27.45 (22.79)	22.25 (11.92)	0.3416
Creatinine (mg/dL)	0.76 (0.44)	0.77 (0.50)	0.75 (0.32)	0.72 (0.22)	0.68 (0.16)	0.0534
Comorbidities, *n* (%)						
Hypertension	309 (10.7)	197 (10.4)	78 (10.6)	23 (14.9)	11 (11.0)	0.3662
Diabetes mellitus type 2	138 (4.8)	77 (4.0)	43 (5.9)	12 (7.8)	6 (6.0)	0.0474
Renal failure	4 (0.1)	3 (0.2)	1 (0.1)	0 (0.0)	0 (0.0)	1.0000
Metabolic syndrome	525 (18.2)	339 (17.8)	136 (18.6)	36 (23.4)	14 (14.0)	0.2503
Coronary artery disease	29 (1.0)	21 (1.1)	6 (0.8)	2 (1.3)	0 (0.0)	0.7713
Monitoring region, *n* (%)						
Northern region	571 (19.8)	312 (16.4)	182 (24.8)	36 (23.4)	41 (41.0)	
Central region	558 (19.3)	390 (20.5)	139 (19.0)	21 (13.6)	8 (8.0)	
Southern region	1760 (60.9)	1200 (63.1)	412 (56.2)	97 (63.0)	51 (51.0)	<0.0001
Environmental factors, mean (SD)						
Temperature (°C)	24.33 (0.77)	24.31 (0.72)	24.41 (0.84)	24.36 (0.93)	24.18 (1.04)	0.0040
Relative humidity (%)	74.29 (2.45)	74.2 (2.47)	74.51 (2.37)	74.22 (2.49)	74.57 (2.47)	0.0157
Rainfall (mm/day)	0.22 (0.05)	0.22 (0.05)	0.21 (0.05)	0.23 (0.05)	0.23 (0.05)	<0.0001
Air pollution factors, mean (SD)						
PM_10_ (μg/m3)	68.07 (17.06)	69.05 (16.99)	65.72 (17.51)	68.5 (14.8)	65.97 (16.56)	<0.0001
PM_2.5_ (μg/m3)	37.65 (10.74)	38.44 (10.74)	35.88 (10.74)	38.36 (10.15)	34.45 (9.56)	<0.0001
CO (ppm)	0.45 (0.18)	0.44 (0.17)	0.45 (0.20)	0.48 (0.19)	0.54 (0.26)	<0.0001
NO (ppb)	4.18 (4.01)	4.00 (3.64)	4.31 (4.29)	4.46 (4.57)	6.21 (6.34)	<0.0001
NO_2_ (ppb)	14.98 (5.65)	14.9 (5.23)	14.76 (6.45)	15.77 (5.33)	16.9 (6.89)	0.0011
NO_X_ (ppb)	19.14 (8.93)	18.88 (8.19)	19.06 (9.94)	20.21 (9.23)	23.09 (12.45)	<0.0001
O_3_ (ppb)	30.91 (3.86)	30.94 (3.78)	31.04 (4.04)	30.71 (3.63)	29.54 (4.20)	0.0028
SO_2_ (ppb)	3.66 (1.20)	3.61 (1.09)	3.70 (1.39)	3.82 (1.20)	4.00 (1.42)	0.0017

Characteristics of the study participants for the continuous and categorical variables were analyzed using one-way ANOVA and the chi-squared/Fisher’s exact test, as appropriate, for comparisons among spirometry groups. SD: Standard deviation; PM_2.5_: Fine particulate matter PM_10_: Coarse particulate matter; CO: Carbon monoxide; NO: Nitrogen monoxide; NO_2_: Nitrogen dioxide; NO_x_: Nitrogen oxides; O_3_: Ozone; SO_2_: Sulfur dioxide.

**Table 2 jpm-11-00819-t002:** Stepwise multinomial logistic regression analysis.

	Obstructive vs. Normal		Restrictive vs. Normal		Mixed vs. Normal	
	Adjusted OR (95% CI)	*p*	Adjusted OR (95% CI)	*p*	Adjusted OR (95% CI)	*p*
Age group (years)						
30–39	1.00		1.00		1.00	
40–49	1.00 (0.77–1.3)	0.9876	1.11 (0.65–1.89)	0.6944	0.65 (0.3–1.42)	0.2809
40–59	0.98 (0.76–1.27)	0.8847	1.08 (0.64–1.81)	0.7836	0.89 (0.44–1.8)	0.7546
≥60	1.15 (0.86–1.54)	0.3385	1.38 (0.80–2.39)	0.2532	2.64 (1.35–5.13)	0.0043
Gender						
Male	1.00		1.00		1.00	
Female	-		-		-	-
Anthropometric parameters						
Height	0.99 (0.98–1.01)	0.2861	0.96 (0.94–0.98)	<0.0001	0.93 (0.91–0.96)	<0.0001
Weight	-		-		-	-
Body Adiposity Index	-		-		-	-
Body Roundness Index	-		-		-	-
Biochemical data						
Hematocrit	-		-		-	-
Glycohemoglobin	1.13 (1.01–1.27)	0.0312	1.28 (1.09–1.51)	0.0028	0.94 (0.69–1.28)	0.7093
Albumin	-		-		-	-
Creatinine	-		-		-	-
Comorbidities						
Diabetes mellitus type 2	-		-		-	-
Monitoring region						
Northern region	1.00		1.00		1.00	
Central region	0.94 (0.59–1.50)	0.7913	0.32 (0.13–0.78)	0.0124	0.07 (0.02–0.24)	<0.0001
Southern region	0.61 (0.39–0.97)	0.0351	0.31 (0.13–0.74)	0.0083	0.14 (0.05–0.44)	0.0007
Meteorological factors						
Temperature	1.58 (1.35–1.86)	<0.0001	1.30 (0.96–1.77)	0.0906	1.87 (1.26–2.77)	0.0020
Relative humidity	1.06 (1.01–1.11)	0.0104	1.07 (0.99–1.16)	0.0870	1.15 (1.03–1.28)	0.0108
Rainfall	-		-		-	-
Air pollution factors						
PM_10_	-		-		-	-
PM_2.5_	0.97 (0.96–0.99)	0.0002	1.00 (0.97–1.03)	0.9868	0.97 (0.93–1.01)	0.0930
CO	1.19 (0.45–3.14)	0.7219	20.3 (3.14–131.12)	0.0016	0.87(0.08–9.77)	0.9122
NO	-		-		-	-
NO_2_	1.01 (0.91–1.13)	0.8136	1.39 (1.14–1.7)	0.0012	1.58 (1.25–1.99)	0.0001
NO_X_	0.98 (0.91–1.06)	0.6126	0.76 (0.66–0.87)	0.0001	0.86 (0.73–1.02)	0.0809
O_3_	-		-		-	-
SO_2_	1.26 (1.09–1.45)	0.0015	0.91 (0.68–1.22)	0.5256	0.80 (0.56–1.13)	0.2058

OR: odds ratio; CI: confidence interval; SD: Standard deviation; CO: Carbon monoxide; NO: Nitrogen monoxide; NO_2_: Nitrogen dioxide; NO_x_: Nitrogen oxides; O_3_: Ozone; SO_2_: Sulfur dioxide; PM_10_: coarse particulate matter; PM_2.5_: fine particulate matter. Data for CO were skewed and log transformed for analysis.

**Table 3 jpm-11-00819-t003:** Interactions between temperature and air pollutants.

	Low Temperature						High Temperature					
	Obstructive vs. Normal		Restrictive vs. Normal		Mixed vs. Normal		Obstructive vs. Normal		Restrictive vs. Normal		Mixed vs. Normal	
	Adjusted OR (95% CI)	*p*	Adjusted OR (95% CI)	*p*	Adjusted OR (95% CI)	*p*	Adjusted OR (95% CI)	*p*	Adjusted OR (95% CI)	*p*	Adjusted OR (95% CI)	*p*
PM_10_	0.99 (0.98–0.99)	0.0084	0.99 (0.97–1.01)	0.3815	0.98 (0.96–1.01)	0.1297	0.99 (0.99–1.00)	0.1169	1.01 (0.99–1.02)	0.1942	1.01 (0.99–1.02)	0.5185
PM_2.5_	0.98 (0.96–0.99)	0.0133	0.97 (0.93–1.00)	0.0845	0.87 (0.82–0.93)	<0.0001	0.98 (0.97–0.99)	0.0015	1.02 (1.00–1.04)	0.0458	1.00 (0.98–1.02)	0.9675
CO	1.21 (0.88–1.66)	0.2421	1.78 (0.98–3.25)	0.0596	5.35 (2.88–9.94)	<0.0001	0.93 (0.63–1.39)	0.7307	4.21 (1.77–9.99)	0.0011	2.07 (0.84–5.11)	0.1163
NO	1.03 (1.01–1.06)	0.0041	1.05 (1.01–1.09)	0.0137	1.11 (1.07–1.15)	<0.0001	1.06 (0.94–1.18)	0.3491	0.86 (0.69–1.06)	0.1604	1.14 (0.87–1.49)	0.3313
NO_2_	1.02 (0.99–1.04)	0.2000	1.04 (0.99–1.09)	0.0659	1.13 (1.08–1.19)	<0.0001	1.00 (0.98–1.03)	0.7627	1.06 (1.01–1.12)	0.0251	1.07 (1.00–1.14)	0.0406
NO_X_	1.01 (1.00–1.03)	0.0312	1.02 (1.00–1.05)	0.0252	1.06 (1.04–1.08)	<0.0001	1.01 (0.98–1.03)	0.6624	1.04 (0.99–1.08)	0.1144	1.05 (0.99–1.11)	0.0582
O_3_	0.88 (0.83–0.93)	0.8177	0.97 (0.92–1.02)	0.2219	0.88 (0.83–0.93)	<0.0001	0.88 (0.75–1.03)	0.0008	0.98 (0.87–1.11)	0.7681	0.88 (0.75–1.03)	0.1116
SO_2_	1.04 (0.92–1.19)	0.5211	1.29 (1.01–1.65)	0.0402	1.77 (1.36–2.30)	<0.0001	1.12 (1.00–1.24)	0.0426	1.17 (0.96–1.41)	0.1189	1.38 (1.08–1.78)	0.0104

Adjusted ORs and 95% CIs were estimated using multinomial logistic regression after categorization by age, height, white blood cells, glycohemoglobin, and relative humidity to delineate the covariant factors associated with the development of lung disease when the factors showed a significant association in the crude analysis. Data for CO were skewed and log transformed for analysis.

**Table 4 jpm-11-00819-t004:** Interactions among monitoring area and air pollutants.

	Northern and Central Region						Southern Region					
	Obstructive vs. Normal		Restrictive vs. Normal		Mixed vs. Normal		Obstructive vs. Normal		Restrictive vs. Normal		Mixed vs. Normal	
	Adjusted OR (95% CI)	*p*	Adjusted OR (95% CI)	*p*	Adjusted OR (95% CI)	*p*	Adjusted OR (95% CI)	*p*	Adjusted OR (95% CI)	*p*	Adjusted OR (95% CI)	*p*
PM_10_	0.99 (0.98–0.00)	0.0769	1.02 (0.99–0.04)	0.0626	1.01 (0.99–1.03)	0.1390	0.99 (0.98–1.00)	0.1266	0.96 (0.94–0.99)	0.0029	0.98 (0.95–1.02)	0.3101
PM_2.5_	0.98(0.96–0.99)	0.0048	1.00 (0.97–0.03)	0.8962	0.98 (0.95–1.01)	0.2379	0.97 (0.96–0.99)	0.0030	1.00 (0.97–1.03)	0.8837	0.97 (0.93–1.02)	0.2641
CO	0.85 (0.66–0.10)	0.2096	1.48 (0.89–0.46)	0.1276	2.42 (1.44–4.06)	0.0008	4.17 (2.08–8.37)	<0.0001	11.58 (3.29–40.81)	0.0001	18.85 (3.53–100.59)	0.0006
NO	1.00 (0.98–0.03)	0.7094	1.03 (0.99–0.08)	0.1062	1.08 (1.04–1.12)	0.0002	1.28 (1.15–1.44)	<0.0001	0.89 (0.70–1.14)	0.3524	1.33 (1.02–1.73)	0.0349
NO_2_	0.99 (0.97–0.00)	0.1122	1.03 (0.99–0.06)	0.1903	1.06 (1.02–1.10)	0.0033	1.13 (1.08–1.18)	<0.0001	1.09 (1.01–1.18)	0.0263	1.25 (1.13–1.37)	<0.0001
NO_X_	1.00 (0.98–0.01)	0.4423	1.02 (0.99–0.04)	0.1333	1.04 (1.02–1.06)	0.0006	1.09 (1.06–1.13)	<0.0001	1.05 (0.99–1.11)	0.1367	1.16 (1.08–1.25)	<0.0001
O_3_	1.01 (0.98–0.04)	0.4920	0.95 (0.90–0.02)	0.1542	0.89 (0.83–0.95)	0.0004	0.99 (0.95–1.03)	0.5165	1.01 (0.94–1.09)	0.7119	0.91 (0.82–1.01)	0.0838
SO_2_	0.95 (0.86–0.05)	0.3577	1.19 (0.99–0.44)	0.0658	1.34 (1.10–1.63)	0.0031	1.65 (1.45–1.89)	<0.0001	1.29 (1.00–1.65)	0.0491	1.83 (1.4–2.38)	<0.0001

Adjusted ORs and 95% CIs were estimated using multinomial logistic regression after categorization by age, height, white blood cells, glycohemoglobin, and relative humidity to delineate the covariant factors associated with the development of lung disease when the factors showed a significant association in the crude analysis. Data for CO were skewed and log transformed for analysis.

## Data Availability

The data underlying this study is from the Taiwan Biobank. Due to restrictions placed on the data by the Personal Information Protection Act of Taiwan, the minimal data set cannot be made publicly available. Data may be available upon request to interested researchers. Please send data requests to: Szu-Chia Chen, Division of Nephrology, Department of Internal Medicine, Kaohsiung Medical University Hospital, Kaohsiung Medical University.

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
