# Peer review of "The Impact of the Synergistic Effect of Temperature and Air Pollutants on Chronic Lung Diseases in Subtropical Taiwan"

_jpm, 2021, doi:10.3390/jpm11080819_

Round 1

Reviewer 1 Report

The manuscript entitled “The impact of the synergistic effect of temperature and air pollutants on chronic lung diseases in subtropical Taiwan” describes the results of a cross-sectional study aimed at investigating the association between chronic lung disease, air pollution, meteorological factors, and anthropometric indices. The study uses a rigorous methodology and produces conclusions regarding the necessity of using temperature in conjunction with pollutants when assessing the impact on respiratory diseases.

The variables considered as confounders of the association between pollutants, meteorological factors and respiratory diseases are socio-demographic variables, biochemical parameters and anthropometric indices. It is well known that in studies of this type it is necessary to assess occupational exposure because the effect of toxic substances can interact with or add to environmental pollutants. The authors should explain why they did not make this assessment and whether the data can be used in the model.

Author Response

We appreciate the reviewer for carefully reading our manuscript and revising it. Thank you very much for your helpful and insightful comments. We agree that occupational exposure to toxic substances can interact with or add to environmental pollutants. The data in this study came from the Taiwan Biobank, which is the first and largest government-supported biobank, that can be used for population-based cohort studies and disease-oriented research. However, in the Taiwan Biobank questionnaire, occupational information only includes current and past job categories, positions, duration, and job content. The questionnaire does not contain specific toxic substance exposure records. Therefore, we cannot analyze occupational exposure to toxic substances in this study. This has been added to the limitation section.

  • “Finally, the Taiwan Biobank does not contain information regarding occupational exposure to toxic substances, and therefore this variable cannot be analyzed in this study.” (Line 456 - 458).

Reviewer 2 Report

The paper of Da-Wei Wu et al researched the synergistic effects of temperature, monitoring area and air pollutants on chronic lung diseases in a big cohort of over 2889 participants. The paper is well written and understandable for a broad public. Some items in the methods and discussion however, needs better explanation. The comments can be found in the attached file. 

Author Response

Introduction

  1. “elderly patients, and that this effect was restricted to the cold season” Please define “elderly patients” and “cold season”

Ans: Thank you for your valuable comments. According to reference [18], which was conducted in Yancheng, China, "elderly patients" refer to people over 75 years of age, and "cold season" refers to the time of the year from November to April. In our study, elderly patients refer to participants over 60 years old, and the cold season refers to the winter period from December to February. The definitions of "elderly patient" and "cold season" have been added to the introduction.

  • “A study reported that exposure to PM2.5 was associated with a higher risk of acute exacerbations of COPD, especially in females and elderly patients (age > 75 years old), and that this effect was restricted to the cold season (November to April) [18].” (Line 102-103)

Methods

  1. Assessment of lung function status: This capital is to short, the procedure should be better explained Which lung function parameters were obtained and evaluated? Was there a standardized protocol? Were beta2-agonists were stopped 12 h before lung function measurements? Who evaluated the validity of all the lung function tests and was he/she blinded to the other outcomes?

   Ans: Thank you for your valuable comments. This is a retrospective study using data from the Taiwan Biobank. The lung function parameters in the Taiwan Biobank include forced expiratory volume in one second (FEV1), forced vital capacity (FVC), FEV1/FVC% ratio, FVC-predicted (or FVC%-predicted) value, and FEV1-predicted (or FEV1%-predicted) value. FVC-predicted and FEV1-predicted values were determined by dividing the measured value by the reference value, which was calculated from a formula derived from the general population based on gender, age, height, and Asian ethnicity.

    The Taiwan Biobank data came from multiple centers. The spirometry measurements were performed by well-trained technicians using the MicroLab spirometer and Spida 5 software (Micro Medical Ltd., Rochester, Kent, UK) [27]. Each participant received 3 pulmonary function tests and the best result from the 3 tests was included, in accordance with the quality standards of the American Thoracic Society guidelines [1]. Most of the participants were healthy adults who had not received β2 agonist treatment, but for those that did receive β2 agonists, the above guidelines were followed. Since this is a retrospective study using cross-sectional lung function data, validation of lung function tests and blinding of technicians could not be performed. We have added more detailed information in section 2.3 Assessment of Pulmonary Function Status.

  • “The lung function parameters in the Taiwan Biobank include forced expiratory volume in one second (FEV1), forced vital capacity (FVC), FEV1/FVC% ratio, FVC-predicted value, and FEV1-predicted value. FVC-predicted and FEV1-predicted values were determined by dividing the measured value by the reference value, which was calculated from a formula derived from the general population based on gender, age, height, and Asian ethnicity. The spirometry measurements were performed by well-trained technicians using the MicroLab spirometer and Spida 5 software (Micro Medical Ltd., Rochester, Kent, UK) [27]. Each participant underwent 3 lung function tests and the best result from the 3 tests was included, in accordance with the quality standards of the American Thoracic Society guidelines. We defined chronic lung diseases in terms of three abnormal types of lung function, including obstructive lung diseases (FEV1/FVC < 70%, such as asthma, chronic obstructive pulmonary disease, and bronchiectasis), restrictive lung diseases (FEV1/FVC > 70% and FVC<80%, such as interstitial lung disease, neuromuscular disease, and obesity), and mixed (obstructive and restrictive) lung disease, according to the standardized European Respiratory Society/American Thoracic AssociationSociety guidelines (Miller et al., 2005) [1].” (Line 147-160)

[27] Hsieh, S.W.; Wu, D.W.; Wang, C.W.; Chen, S.C.; Hung, C.H.; Kuo,

C.H. Poor cognitive function is associated with obstructive lung

diseases in Taiwanese adults. Int J Environ Res Public Health. 2021, 18,

  1. [CrossRef]

Statistical analysis

  1. Please explain: Were the variables used for the Pearson correlation parametric? Which test was used to define that?

   Ans: Thank you for your comments and valuable suggestions. In this study, normality of data was assessed via the Kolmogorov-Smirnov test. Since Kolmogorov-Smirnov D < 0.01, this rejects the null hypothesis that the variable is normally distributed at the 0.05 level. Due to the non-normal distribution of the data, we have used Spearman's rank correlation instead of Pearson product-moment correlation to analyse the correlation between air pollutants (PM10, PM2.5, CO, NO, NO2, NOx, O3, SO2) and meteorological (temperature, relative humidity, rainfall) factors (Table S2). As expected from the large-sample normal approximation theory, the results from the Spearman's rank correlation and the Pearson product-moment correlation were similar. We have revised the results in Section 3.2 Correlation between meteorological factors and outdoor air pollutants.

  • Pearson product-momentSpearman's rank correlation was used to” (Line 172)
  • “We found that Ttemperature was positively correlated with PM5, PM10, O3, and SO2 concentrations. Relative humidity was negatively correlated with all other meteorological factors and air pollutants except for temperature and O3 level. PM2.5 and PM10 concentration werewas positively correlated with all meteorological factors except for relative humidity, and all air pollutants except for O3 and NO. Except for PM2.5 and PM10, CO had very high correlations with all air pollutants (especially NO, NO2, and NOx). In addition, there was a strongmoderate negative correlation between CO and O3. (Table S23).” (Line 213-220)

Results

  1. 1 Descriptive statistics of the demographic, laboratory, meteorological factors and air pollutants Please change ‘olderly’ for ‘elder’ or better define (from which age?) In addition the sentence “However, we found that olderly women who were lower body weight and height, higher BAI and BRI, relative anemia, hypoalbuminemia, and with probable diabetes had higher risk of chronic lung disease” is not very clear.

Ans: Thank you for your correction and valuable comment. We have rewritten this paragraph.  

  • However, we found that olderly women who were lower body weight and height, higher BAI and BRI, relative anemia, hypoalbuminemia, and with probable diabetes had higher risk of chronic lung disease. In addition, when compared with the normal spirometry group, we found that factors associated with higher risk of chronic lung diseases include elderly age (>60 years), female gender, lower body height and weight, higher body adiposity index and body roundness index, lower hematocrit, higher glycohemoglobin, and lower albumin level (Table 1). These results were confirmed by post hoc analysis (Table S1).” (Line 193-200).

Discussion

  1. There are some sentences and points which need to be better explained/defined:

“In this analysis of 2,889 participants registered in the TWB, we found differences in the sex and age of the study population, and that elderly women with lower body weight and height, higher BAI and BRI, relative anemia, hypoalbuminemia, and with probable diabetes” Is not very clear

Ans: Thank you for your correction and comment. We have corrected the error and rewritten this paragraph.

  • “In this analysis of 2,889 participants registered in the TWB, we found differences in the sex and age of the study population, and that elderly women with lower body weight and height, higher BAI and BRI, relative anemia, hypoalbuminemia, and with probable diabetes were more likely to suffer from chronic lung diseases, especially in mixed lung disease that factors associated with higher risk of chronic lung diseases include elderly age (>60 years), female gender, lower body height and weight, higher body adiposity index and body roundness index, lower hematocrit, higher glycohemoglobin, and lower albumin level.” (Line 322-329)
  • “The first important finding of this study was that the older women with a relatively lower body weight and height, and high BAI and BRI, relative anemia, hypoalbuminemia, and with probable diabetes elderly age (>60 years), female gender, lower body height and weight, higher body adiposity index and body roundness index, lower hematocrit, higher glycohemoglobin, and lower albumin level had the higher risk for chronic lung diseases. ” (Line 336-341)

  1. Please define elderly in “elderly women”, define lower body weight and height. Lower body weight but higher BAI? Please explain.

Ans: Thank you for your valuable comments. “Elderly women” refers to women with age > 60 years, “lower body weight and height” refer to the comparison of weight and height between the normal spirometry group and the chronic lung diseases groups. Please see the new Table S1 for more information on the post hoc analysis. In addition, “lower body weight but higher BAI” means that the weight is low, but the body adipose tissue content is high. For example, normal weight obesity (NWO) is a term used to describe patients with higher body adiposity index but normal body weight and body mass index (BMI). NWO is defined as BMI <25 kg/m2 and body fat >30%. Individuals with a normal BMI but a higher proportion of body fat may have increased risk of developing cardiometabolic dysfunction, metabolic syndrome, and higher mortality [28]. Furthermore, a study of cystic fibrosis patients showed that higher adiposity in these patients is associated with restrictive pulmonary dysfunction [29]. Therefore, it is possible that people with low or normal weight but high BAI may have difficulty breathing due to poor lung function and cardiometabolic dysfunction.

[28] Oliveros, E.; Somers, V.; Sochor, O.; Goel, K.; Lopez-Jimenez, F. The

concept of normal weight obesity. Prog Cardiovasc Dis. 2014, 56, 426-33

[CrossRef].

[29] Alvarez, J.A.; Ziegler, T.R.; Millson, E.C.; Stecenko, A.A. Body composition

and lung function in cystic fibrosis and their association with adiposity and normal-weight obesity. Nutrition. 2016, 32, 447-52 [CrossRef].

We have added further explanations in this paragraph.

  • “In addition, lower body weight and height but higher BAI means that the weight is low, but the body adipose tissue content is high. For example, normal weight obesity (NWO) is a term used to describe patients with higher body adiposity index but normal body weight and body mass index (BMI) [28,29].” (Line 341-344).
  1. “The Towards a Revolution in COPD Health (TORCH) study showed that high-risk factors for moderate and severe COPD exacerbations were older age, lower body mass index, and female with poor baseline lung function” This needs further explanation.

Ans: Thank you for your comment. Reference [2730] is a multicenter, randomized, double-blind, parallel group, 3-yr study conducted in 42 countries around the world. High-risk factors for moderate and severe COPD exacerbations were found to be older age (patients ³75 yrs of age had 20% more exacerbations compared with patients < 55 yrs of age), lower body mass index (patients with a BMI ³ 29 kg/m2 had 10% fewer exacerbations compared with patients with BMI of 20 to <25 kg/m2 at baseline, while a low BMI < 20 kg/m2 was associated with 10% increased rate of exacerbations), and females with poor baseline lung function (males had 30% lower exacerbation rate than females). The definitions of "older age", "lower body mass index" and "female with poor baseline lung function" have been added to the 2nd paragraph of discussion.

  • “The Towards a Revolution in COPD Health (TORCH) study showed that high-risk factors for moderate and severe COPD exacerbations were older age (patients ³75 years of age), lower body mass index (BMI < 20 kg/m2), and females with poor baseline lung function (females had 1.42 times higher exacerbation rate than males) [2730]” (Line 346-348).
  1. Obesity is in generally associated with decreased lung-function (Int J Chron Obstruct Pulmon Dis.2014; 9: 723–733.), how was this in the cohort? Perhaps you mean lower body mass index (lower than…) or higher than….?

Ans: Thank you for your valuable comments and suggestions. Obesity is one of the major causes of morbidity and mortality worldwide, and has been associated with decreased lung function in epidemiologic studies [a]. However, there are two traditional classifications of COPD: obese type "fat blue" and thin type "pink puffer", which means that being either overweight or underweight may be associated with poor lung function [b]. In addition, according to the definition of the World Health Organization (WHO), overweight refers to a BMI greater than or equal to 25 kg/m2, and obesity refers to a BMI greater than or equal to 30 kg/m2. In our cohort, a total of 2889 participants had an average BMI of 24.12 kg/m2. (Table 1). The average BMI value of the mixed lung disease group was lower (23.38 kg/m2), but it was still in the upper range of normal. This means that no association was found between obesity (BMI ³ 30 kg/m2) and poor lung function in this cohort.

[a] Hanson, C.; Rutten, E.P.; Wouters, E.F.; Rennard, S. Influence of diet and

obesity on COPD development and outcomes. Int J Chron Obstruct Pulmon

Dis. 2014, 9, 723-33. [CrossRef]

[b] Spelta, F.; Fratta Pasini, A.M.; Cazzoletti, L.; Ferrari, M. Body weight and

mortality in COPD: focus on the obesity paradox. Eat Weight Disord. 2018, 23, 15-22. [CrossRef]

  1. “However, the other air pollutants (O3, PM5, and PM10) were not correlated with the incidence of chronic lung diseases” please explain further and compare with other study. Is the exposition not high or not chronical enough?

Ans: Thank you for your valuable comment. We have already discussed this issue in the manuscript: “O3 had the opposite effect to other air pollutants (such as NO2x) in this study, which is consistent with a previous study [5357]. A possible explanation is that NO2x and volatile organic compounds react and combine under ultraviolet light to form ground-level O3. Therefore, when the NO2x concentration drops, the O3 concentration will gradually increase. Although there was no obvious relationship between PM2.5, PM10 and chronic lung diseases in this study, a previous study had pointed out that the concentration of PM increases with temperature, and thus the impact of PM2.5 on lung diseases may only occur at higher temperatures [5459]. Our data is consistent with the result of the above study.” (Line 424-435). We have also added some references in this paragraph. 

  • “The average level of O3 in this study was lower than the world average [56], and therefore its impact on lung diseases might not be obvious. In addition, O3 levels had the opposite effect to other air pollutants (such as NO2x) in this study, which is consistent with a previous study [5357]. A possible explanation is that NO2x and volatile organic compounds react and combine under ultraviolet light to form ground-level O3 [58]. Therefore, when the NO2x concentration drops, the O3 concentration will gradually increase. Although in this study, there was no obvious relationship between PM5, PM10 and chronic lung diseases in this study, a previous study had pointed out that the concentration of PM increases with temperature, and thus the impact of PM2.5 on lung diseases may only occur at higher temperatures [5459]. In addition, a study showed that hospital admission rates for respiratory diseases increased with increasing PM and temperature [60]. Our data were consistent with the results of the above studies. Furthermore, it is possible that the lack of association between PM2.5 and PM10 and the incidence of chronic lung diseases in this study may be due to the level of exposure not being high enough or the duration of exposure not being long enough.” (Line 423-437).

[56] Malley, C.S.; Henze, D.K.; Kuylenstierna, J.C.I.; Vallack, H,W.; Davila

Y, Anenberg, S.C.; Turner, M.C.; Ashmore, M.R. Updated Global

Estimates of Respiratory Mortality in Adults ≥30Years of Age

Attributable to Long-Term Ozone Exposure. Environ Health Perspect.

2017, 125, 087021. [CrossRef]

[58] Zhang, J.J; Wei, Y.; Fang, Z. Ozone Pollution: A Major Health Hazard

Worldwide. Front Immunol. 2019, 10, 2518. [CrossRef]

[60] Jo, E.J.; Lee, W.S.; Jo, H.Y.; Kim, C.H.; Eom, J.S.; Mok, J.H.; Kim,

M.H.; Lee, K.; Kim, K.U.; Lee, M.K.; Park, H.K. Effects of particulate

matter on respiratory disease and the impact of meteorological factors in

Busan, Korea. Respir Med. 2017, 124, 79-87. [CrossRef]

  1. “Although previous studies have shown that low concentrations of CO can be used as a treatment option for COPD” not sure if this is relevant since its rather known as an inhalation hazard, as far as I know therapies are rather experimental.

Ans: Thank you for your comment and correction. We have modified this paragraph.

  • “Although previous studies have shown that low concentrations of CO can be used as a treatment option for COPD, this method is still experimental and not routinely used in the clinical setting [4245]” (Line 397-399).
  1. “Almost all air pollutants showed higher concentrations in southern Taiwan, where the temperature is also higher.” Are there other explanations which are excluded? For example industries, more dense population and cars? In the conclusion you mention southern Taiwan as “heavy industrial regions”.

Ans: Thank you for your comment and correction. We have modified this sentence.

  • “Almost all air pollutants showed higher concentrations in southern Taiwan, where the temperature is also higher, and where there is a higher density of industrial factories (there are more than 5,000 factories in Kaohsiung City).” (Line 381-383).
  1. It should be mentioned in the discussion that the lung function can also be influenced by other (not measured) harmful pollutants, for example systematically or by inhalation (for example flame retardants in indoor air (brominated diphenylethers (BDEs), polychlorinated biphenyls). Therefore, the sentence “In addition, we did not have information regarding indoor air quality.” Should be a bit more outlined.

Ans: Thank you for your valuable comment. We have revised this section.

  • “In addition, we did not have information regarding indoor air quality. Lung function can also be influenced by exposure to indoor air pollutants (such as brominated diphenylethers, polychlorinated biphenyls [61], volatile organic compounds, polycyclic aromatic hydrocarbons, PM5, cigarette smoke, and cooking) [62]. This will be an important topic for further study.” (Line 452-456).

[61] Zhang, X.; Diamond, M.L.; Robson, M.; Harrad, S. Sources, emissions,

and fate of polybrominated diphenyl ethers and polychlorinated biphenyls indoors in Toronto, Canada. Environ Sci Technol. 2011, 45, 3268-3274. [CrossRef]

[62] Vardoulakis, S.; Giagloglou, E.; Steinle, S.; Davis, A.; Sleeuwenhoek, A.;  

Galea, K.S.; Dixon, K.; Crawford, J.O. Indoor Exposure to Selected Air

Pollutants in the Home Environment: A Systematic Review. Int J Environ Res Public Health. 2020, 17, 8972. [CrossRef]

Conclusion

  1. “In conclusion, we found that elderly women who were lower body weight and height, higher BAI and BRI, relative anemia, hypoalbuminemia, and with probable diabetes had higher risk of chronic lung disease, especially in mixed lung disease when exposed to air pollution.” Is not very clear, see comments above.

Ans: Thank you for your comment and suggestion. We have revised this section.

  • In conclusion, we found that elderly women who were lower body weight and height, higher BAI and BRI, relative anemia, hypoalbuminemia, and with probable diabetes had higher risk of chronic lung disease, especially in mixed lung disease when exposed to air pollution when compared with the normal spirometry group, we found that factors associated with higher risk of chronic lung diseases include elderly age (>60 years), female gender, lower body weight and height, higher BAI and BRI, lower haematocrit, lower albumin level, and higher glycohemoglobin level (Table 1).” (Line 459-465).
